# Use of Immunochromatographic SARS-CoV-2 Antigen Testing in Eight Long-Term Care Facilities for the Elderly

**DOI:** 10.3390/healthcare9070868

**Published:** 2021-07-09

**Authors:** Martina Ifko, Miha Skvarc

**Affiliations:** 1Doba Faculty, 2000 Maribor, Slovenia; 2Medical Faculty Ljubljana, University of Ljubljana, 1000 Ljubljana, Slovenia; miha.skvarc@sb-je.si; 3General Hospital Jesenice, 4270 Jesenice, Slovenia

**Keywords:** immunochromatographic SARS-CoV-2 antigen test, clinical validation, POCT—point of care testing, long-term care

## Abstract

The clinical validation of the NADAL COVID-19 antigen test (Nal von Minden, Moers, Germany) started in eight Slovenian long-term health care facilities in October 2020. The purpose of clinical validation is to implement the test into the everyday working process in long-term care (LTC) facilities and demonstrate how it can be used to mitigate the spread of the virus in these environments. The facilities compared the results of antigen tests to the results obtained using Cobas 6800 SARS-CoV-2 real-time reverse transcription polymerase chain reaction (RT-PCR) (Roche, USA). Sensitivity (86.96%, 95% CI: 66.41–97.23%) and specificity (88.24%, 95% CI: 80.35–93.77%) of the NADAL COVID-19 antigen test were good. Rapid antigen testing served well for early detection of infection and helped to prevent and control spread of the SARS Cov2 in six out of eight LTCs. Moreover, mini-outbreaks were quickly resolved in all six LTCs. Locally validated immunochromatographic SARS-CoV-2 antigen testing can be used to contain the spread of the virus in LTCs. Antigen tests also deliver accurate information very quickly if used early with a low threshold. The NADAL COVID-19 antigen test proved to be a good screening tool to detect SARS-COV-2 in LTCs.

## 1. Introduction

The COVID-19 pandemic continues to pose a major threat to public health in European countries. On 28 October 2020, European Commission Communication stated that “robust testing strategies and sufficient testing capacities are essential aspects of preparedness and response to COVID-19” [1]. Rapid antigen tests can give quick results and can be used massively by efficiently trained non-laboratory health care personnel as point-of-care tests (POCTs). Such results could be used as a mitigation step to stop the uncontrolled spreading of the virus in hospitals [2]. Long-term care (LTC) facilities for the elderly are high-risk settings for transmission of severe acute respiratory syndrome coronavirus 2 (SARS-CoV-2), the causative agent of coronavirus disease 2019 (COVID-19) [3]. In Slovenia, more than 80% of hospitalized cases originated from LTCs. In majority of cases, the virus was brought to the LTCs by the facility staff. Given the high mortality rate associated with COVID-19 among LTC residents, we believe that timely and evidence-based methods like the use of antigen tests can be critical to curbing the viral spread in these settings. In late October 2020, Slovenia had one of the highest burdens of COVID-19 in the world. LTCs had to wait for a long time, i.e., more than 24 h, to receive RT-PCR results and because the 14-day incidence was above 1000 per 100,000 inhabitants and the pressure on health care institutions was extreme [4], LTC management teams decided to use antigen testing as a screening tool to detect infection in their employees and residents. The purpose of the report is to present the use of SARS-CoV-2 antigen tests for LTC residents and staff to actively prevent outbreaks in eight LTCs for the elderly. The first part of the report a is validation study, while the second part is a description of routine use of antigen tests to prevent outbreaks in the LTCs. The study protocol was approved by the Institutional ethics committees (No. 2020-02).

## 2. Materials and Methods

Eight LTCs started using NADAL COVID-19 (Nal von Minden, Moers, Germany) on 15 October 2020 with the purpose to prevent the spread of the virus among LTC residents. The eight LTCs included in the survey take care for a total of 1237 residents, with the mean number of beds amounting to 155. The biggest LTC included in the survey has 205 beds. Elderly age group included in the study was 65–95 years. Under the supervision of a clinical microbiologist, the LTC staff validated the test against the Cobas 6800 RT-PCR test (Roche, Mannheim, Germany) as suggested by authorities [5]. All LTCs followed the same algorithm while validating the antigen test. In first part of the study they tested residents with at least one sign of COVID-19, e.g., fever above 37.5 °C, felling tired, runny nose, dry cough, dyspnoea, loss of taste and smell or gastrointestinal problems, with at least one symptom of an upper respiratory tract infection. In the second part of the study also pre-symptomatic employees were included in the study and elderly with high risk contact. Two swabs were taken by a trained healthcare provider. First, swab was immediately used to perform antigen test on side of the LTC, following instruction for use of the test. Collected sample was immediately put in an extraction tube, with extraction buffer for two minutes. After extraction was completed two drops of extracted specimen were placed on the test cassette, results were visually read after 15 min. A second and contralateral nasopharyngeal swab was taken by the same patient and by the same health care worker to be used for the RT-PCR. RT-PCR was run by a dedicated team of laboratory technicians in the local public health microbiology laboratory. People who had showed symptoms and signs of COVID-19 for more than five days were excluded from the antigen testing. To optimize performance, the antigen test was performed by trained operators in all institutions. For statistical analysis, we used MedCalc Statistical Software version 18.11.6 (MedCalc Software bvba, Ostend, Belgium; https://www.medcalc.org (accessed on 12 May 2021); 2019). We calculated diagnostic accuracy of the test and 95% confidence interval (95% CI).

The study protocol was approved by the Institutional Review Board of General Hospital Jesenice (Nr. 2020-02) for all LTCs. All study participants for local clinical validation were enrolled by dedicated health care personnel and provided an oral informed consent.

## 3. Results

Table 1 presents the clinical validation data for the NADAL COVID-19 test administered in the local environment on 125 nasopharyngeal swabs. The antigen test proved to be sufficiently accurate to be used in specific settings of LTCs as POCT with very good diagnostic accuracy presented as area under the curve (AUC) and 95% confidence interval: 95% CI (88%, 95% CI: 81–93%). However, the LTCs decided to confirm the antigen test results with an RT-PCR test in the case when a false negative or false positive error was suspected. 

After the validation of the test, LTCs performed the test on all employees who reported feeling ill before they were allowed to start their work. The elderly were tested if they had fever and at least one other sign of COVID-19. A subject was classified as presymptomatic if no symptoms were present for 14 days before a positive SARS-CoV-2 test, but typical or atypical symptoms developed during follow-up; when no symptoms developed during follow-up, the subject was classified as asymptomatic. If an elderly or employee tested positive by antigen test, a sample taken from contralateral nares was sent to RT-PCR testing. In addition, all other residents of the same LTC tract were tested. Other staff who worked in the same shift as the infected worker were also tested for SARS-CoV-2 using the NADAL COVID-19 antigen test. An additional 613 antigen tests were performed by the end of November in seven facilities. Some of the test subjects were also tested by RT-PCR. The antigen test proved to be sufficiently accurate to be used in specific settings of LTCs as POCT with very good diagnostic accuracy employees and residents (area under the curve—AUC for employees 95%, 95% CI: 91–97%, AUC for residents 91%, 95CI: 88–95%). Data are presented in Table 2 and Table 3. Quite a few employees (*n* = 55) and residents (*n* = 53) were tested more than once in the observed period. The administered antigen tests were positive for 3.75% (9) employees and 6.97% (26) residents. RT-PCR testing detected another 2.92% (7) infected employees and 1.61% (6) residents proved to be false positive in the antigen test. Only 0.42% (1) of the employees who were tested two times was tested positive on antigen tests and another one on an RT-PCR test. 11.61% (6) residents tested two times were positive on antigen tests and another 2.14% (8) tested positive on RT-PCR tests. None of the employees or residents who were tested more than two times were positive on the antigen test or on the RT-PCR test.

Through the use of antigen tests, six of the eight monitored LTCs prevented major uncontrolled outbreaks of COVID-19. The 1st LTC that was unable to control the outbreak has 154 residents and is located in the north-central part of Slovenia (data not presented in Table 2). In that facility, the virus was introduced by a mobile resident and resulted in 37 hospitalizations of other residents. A total of 22 employees were also infected (Table 4). They most probably became infected from the residents. Employees were most likely the source of infection for other residents, except for two residents who got infected from the index case. In the 2nd LTC where the outbreak happened (data not presented in Table 2), an aromatherapy session conducted by a member of the LTC staff for the residents resulted in an outbreak among residents. Fortunately, the viral load identified among the residents was not so high and majority of residents were mildly ill or asymptomatic.

## 4. Discussion

This is the first study that compared the use of immunochromatographic SARS-CoV-2 antigen tests and RT-PCR tests in LTCs and used the antigen test as a mitigation procedure to prevent uncontrolled spread of the virus in LTCs. The sensitivity of the test was very good in all environments. The expected drop of sensitivity in real life situations happened in the cases of infected employees that were presymptomatic or had very mild symptoms and a low viral load on their mucosal barrier.

The results of the study are comparable to other studies. In meta-analysis antigen tests, sensitivity varied considerably across five studies on 943 samples (from 0% to 94%). The highest-performing test recorded sensitivity of 89% [6]. One study compared two antigen tests that uses analyser as detection method to single gene direct SARS-CoV-2 RT-PCR. Assays showed a high degree of agreement for SARS-CoV-2 detection as compared with direct single gene RT-PCR [7]. Another study compared several immunochromatographic antigen tests to the RT-PCR. Sensitivity was between 16% and 85%. The study explained the difference through the use of non-validated sample material [8]. In one study, the antigen test identified 70.6% of RT-PCR positive samples. A major limit of the study was that the sample was diluted in transport media and the comparison was conducted off-site [9]. Our study used fresh samples as point-of-care test, which explains the much higher results in comparison to other studies which used stored frozen samples to conduct the validation in an off-site laboratory. Furthermore, the FindDx platform published several validations of the antigen SARS-CoV-2 assays. Additonal study done in hospital settings shows overall sensitivity at 79.6% (95% CI 67.0–88.8%) [10]. Our test showed similar performances to the best SARS-CoV-2 Ag immunochromatographic tests in other comparisons [11,12].

There have only been a few studies conducted in LTCs or similar institutions. In colleges and campuses, SARS COV 2 antigen assay did not to prove to be the tool to prevent outbreaks [13]. In the LTCs, SARS-CoV-2 viral genomes from employees and residents were clustered, suggesting facility-based transmission. It was suggested that LTCs should conduct serial testing of residents and employees, maximize employees testing participation, ensure availability of personal protective equipment and enhance mitigation practices [14].

To prevent a rapid and widespread transmission of SARS-CoV-2 in nursing facilities where even more than half of residents who has tested positive were asymptomatic at the time of testing and most likely contributed to the transmission [15], a more antigen-based testing strategies focused on pre-symptomatic residents and employees that could prevent transmission after SARS-CoV-2 introduction into the facility, as seen in our report. In one facility, the source of the infection was the resident who got infected at the hospital. The outbreak was kept under control by swabbing all residents in the ward, as well as employees that were in contact with the index resident, which led to the discovery and immediate isolation of a total of four asymptomatic employees. In the three LTCs, the source of infection of the residents were the facility employees. However, the outbreak was quickly prevented through timely isolation. The number of infected residents was higher than that of infected staff. All residents in these LTCs stayed in the facility and were mildly ill or even asymptomatic. In the last two LTCs, the number of infected employees was higher than the number of residents. In both LTCs, they had sporadic cases of infected personnel that were immediately isolated.

Rapid and accurate results are extremely important in LTCs, considering that nursing homes were possibly the most vulnerable institutions already before the COVID-19 pandemic and the elderly with pre-existing conditions had a significantly higher risk of severe disease and death [3]. Our study proved that reliable antigen tests if used frequently can be a got mitigation procedure to stop the spread of SARS-CoV-2 in LTC institutions for the elderly. Similar was proposed in early autumn 2020. Frequent testing with antigen tests in LTCs can stop COVID-19 spreading, even if their analytic and clinical sensitivities are inferior to RT-PCR [16].

On the other hand, our report has certain limitations. As not all residents were swabbed for RT-PCR testing, there is a possibility that some cases of asymptomatic infections might have been missed. The validation study in LTCs began in highly prevalent SARS-CoV-2 conditions, resulting in very good sensitivity of the test that may not apply to low-prevalence situations. We were unable to pinpoint the exact index cases in all LTCs.

## 5. Conclusions

Our results show that validated SARS-CoV-2 antigen test fulfil the criteria as defined by World Health Organization with 80% sensitivity and 97% specificity in different LTCs [5]. We had some false negative cases that were explicable with mild disease. If we suspect that the antigen test is a false negative and a patient has symptoms of infection, an RT-PCR test needs to be conducted. Residents of LTCs benefited from the use of SARS-COV-2 antigen tests, since the need for hospitalizations only arose in one uncontrolled outbreak. Use of RT-PCR has added value in those cases when false negative cases antigen test results in high prevalent SARS-COV-2 environment are suspected.

## Figures and Tables

**Table 1 healthcare-09-00868-t001:** NADAL Covid19 antigen assay versus RT-PCR.

	Number	NADAL COVID-19 Ag Assay vs. RT-PCR
Sensitivity	20/23	86.96% 95% CI: 66.41–97.23%
Specificity	90/102	88.24% 95% CI: 80.35–93.77%
Area under the curve	n.a.	88% 95% CI: 81–93%
Positive predictive value	20/32	62.5% 95% CI: 48.91–74.37%
Negative predictive value	90/93	96.77% 95% CI: 91.24–96.86%

CI: Confidence interval; n.a.: Not applicable.

**Table 2 healthcare-09-00868-t002:** Testing of employees in six LTCs (excluding the two LTCs with an uncontrolled outbreak) *.

LTCs Testing Employees	Number	Positive	Negative
Number of antigen tests in employees	240	9	231
Number of employees tested by antigen tests more than once	38	1	72
Number of RT-PCRs in employees	155	16	139
Number of employees tested by RT-PCR more than once	17	1	34

* Employees sensitivity 56.25%, 95% CI: 29.9–80.2%.

**Table 3 healthcare-09-00868-t003:** Testing of residents in six LTCs (excluding the two LTCs with an uncontrolled outbreak) *.

LTCs Testing Residents	Number	Positive	Negative
Number of antigen tests in residents	373	26	347
Number of residents tested by antigen tests more than once	50	8	100
Number of RT-PCRs in residents	216	20	196
Number of residents tested by RT-PCR more than once	13	6	20

* Residents’ specificity 90.7%, 95% CI: 86.1–94.3%.

**Table 4 healthcare-09-00868-t004:** NADAL Covid19 antigen assay versus RT-PCR in asymptomatic employees.

	Number	NADAL COVID-19 Ag Assay vs. RT-PCR
Sensitivity	9/16	56.25% 95% CI:29.87–80.24%
Specificity	139/139	100.00% 95% CI: 97.38–100.00%
Positive predictive value	9/9	100%
Negative predictive value	139/146	95.20% 95% CI: 91.93–97.19%

## Data Availability

The data presented in this study are available on request from the corresponding author. The data are not publicly available due to Slovenian patient right act restriction.

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
