# Peer review of "Use of Immunochromatographic SARS-CoV-2 Antigen Testing in Eight Long-Term Care Facilities for the Elderly"

_healthcare, 2021, doi:10.3390/healthcare9070868_

Round 1

Reviewer 1 Report

Thanks for accepting the most comments and suggestions. Please see few more minors below;

  1. In abstract, please correct the "infection" spelling. 
  2. For the first time please mention the full form of RT-PCR.
  3. In conclusion please correct the "false" spelling.
  4. Please state high and low viral load in the study.

Importantly, I would encourage the authors to provide either in detail supplementary data of RT-PCR as per MIQE guidelines or RT-PCR CT values in specimens tested could be represented as Dot plot. Additionally, results of all Ag tests done can be presented as HeatMap (or any other suitable form like). The Receiver operating characteristic (ROC) curve analysis should be done for the Ag test (Please use this reference paper). These representations will highly encourage a better viewing of the results. 

Author Response

  1. Thank you for pointing this out. We have corrected it.
  2. Thank you for pointing this out. We have added the full form of RT-PCR.
  3. Thank you for pointing this out. We have corrected it.
  4. Thank you for pointing this out. We are aware that this is a drawback of our study but since the end of the year 2020 Slovenian Ministry of Health as well as WHO didn't request that CT values are part of the laboratory report, we do not have these values and we can not collect his information.

Reviewer 2 Report

This manuscript was revised well accordingly. Long-term care results are important for the COVID-19 study. 

  1. The approvement of Ethics Committee should be included in the manuscript.
  2. When and where did authors get the swabs? Please clarify it for better understanding.
  3. Did authors compared the test strip between this manuscript and other reports? 

Author Response

  1. Thank you for pointing this out. We have included the number of approval of Ethics committee.
  2. Thank you for pointing this out. We have included missing information in the Materials and Methods section.
  3. No, we didn't, but we are aware of two publications that studied the sensitivity of the Covid-19 rapid test from Nal Von Minden, where sensitivity and specificity results were similar to our results.

Reviewer 3 Report

The manuscript is really interesting, but a lot of information needed are missed and results are presented in a rather confusing way.

It seems that the paper describes two different studies: a first one focussed on a in-field validation of the rapid antigenic test by comparison with the RT-PCR; a second one describing the use of the rapid antigenic test in the management of the infection in LTC facilities.

Regarding the first study I have the following concerns, comments and requests of amendments:

  • Could the authors provide more details about the characteristics of the rapid antigenic kit and RT-PCR tests used? This info will be extremely useful to properly interpret the results of the study and would explain some affirmations like the one in M&M: “People who had showed symptoms and signs of COVID-19 for more than five days were excluded from the antigen testing, according to the test manufacturer’s instructions for use.”
  • Can the authors confirm that in this first study only residents “with at least one sign of COVID-19” (as reported in the text)? If yes, could the authors explain why this validation has been made only on symptomatic people? As correctly stated by the same authors the pre-symptomatic and a-symptomatic infected persons are the most dangerous for infection spread and those that must be early detected in a community context, like that of LTC. The validation on symptomatic people only may introduce a bias in the results. This should be also properly addressed in the discussion.
  • Another limitation of this validation study is related to the fact that CT values from RT-PCR tests are not reported and considered in the comparison. Could these values be retrieved by the authors and properly analysed in comparison to the positive/negative results by the rapid antigenic test? In fact, it is rather demonstrated that the lower sensitivity of antigenic tests in comparison with molecular assays is linked with swabs with CT values more than 30-35, with low levels of viral genome and possibly with no or very low live viral load.
  • In Table 1 I guess that the right numbers for positive predictive value is 20/32 and not 20/23.

 Regarding the second study (or the other part of the article) I am quite confused. This part describes the use of rapid test in the LTC, but which is the objective? To confirm the sensitivity and specificity values observed in the first study? I do not think that a proper study design was used for that. On the other hand, in order to provide some info on the usefulness of rapid antigenic test use in LTC facilities, it would be useful to understand how many infected people were identified (and how many missed) by the use of this rapid test. And also the proportion of false positive.

However, the diagnostic protocol used in the LTC should be proper described in M&M. Actually this part is completely missing. I guess that not all people has been tested by both tests (antigenic and RT-PCR), and also the reasons for testing are not clearly reported. It is reported that “The elderly were tested if they had fever and at least one other sign of COVID-19” and “LTCs performed the test on all employees who reported feeling ill before they were allowed to start their work”, but then it is reported “ A subject was classified as pre-symptomatic if no symptoms were present for 14 days before a positive SARS-CoV-2 test, but typical or atypical symptoms developed during follow-up; when no symptoms developed during follow-up, the subject was classified as asymptomatic.”. How is it possible? Apparently only people with some symptoms have been tested (see the first two affirmations above). Please clarify in a proper way and in details the diagnostic protocol used in the LTCs and which kind of persons have been tested. This must be reported in the M&M chapter (not in Results).

Table 2 and 3 are really confusing. Since this is not a test validation study, it would be more interesting to report the proportion of infected people corrected identified by the rapid antigenic test.

The last part of the text in Results (after the table 3) is reporting consideration more than results. It should be moved to Discussion.

In Discussion you affirm that “The expected drop of sensitivity in real life situations happened in the cases of infected employees that were presymptomatic or had very mild symptoms and a low viral load on their mucosal barrier”. How can you infer this from your data? I do not think that you reported results supporting this conclusion.

Finally, please clarify whether the papers mentioned in the Discussion chapter, related to other validation studies on antigenic tests, refer to rapid (point-of-care) or laboratory tests.

Author Response

  1. Thank you for pointing this out. We have rewritten this part.
  2. In the time when we started using rapid tests (15th of October 2020), there was only one test available in the market with very low quantities. So we started our validation only on the symptomatic patients that needed to be isolated immediately (please note that at that time, time from sample to result for PCR was 24hour or even more in some cases). That is why we included in the second part of the study (when enough tests were available) also asymptomatic workers, to study if antigen tests can prevent the uncontrolled spread of the disease.
  3. thank you for pointing this out. We are aware that the drawback certainly is that we do not have CT values, but at the time when this data was collected CT values were not reported (this was also not a request from WHO and the local Ministry of health), so this data can not be collected.
  4. Thank you for pointing this out. We have corrected it.
  5. Thank you for pointing this out. We have mentioned the sensitivity of the test under the table. And from the specificity it is seen that rapid test has higher sensitivity in symptomatic patients.
  6. Thank you for pointing this out. The elderly were tested if they had at least one symptom of Covid - 19 or were in high-risk contact (A person who was in a closed environment (e.g. classroom, meeting room, hospital waiting room, etc.) with a COVID-19 case for 15 minutes or more and at a distance of less than 2 metres) with SARS COV 2 positive employee or resident. The employees were tested if they had at least one sign of Covid and had a high-risk contact either at work or at home. Employees were not allowed to work till getting antigen results or if it was assumed that the antigen test was the false-negative, rt-PCR result (in October 2020) it lasted at least 24 hours to get rt-PCR results. We have included this information in the Material and Method section.
  7. Thank you for pointing this out. We have added a table with the requested data.
  8. Thank you for pointing this out. We have moved this part to discussion.
  9. Thank you for pointing this out. We have rewritten this part.
  10. Thank you for pointing this out. Yes, we refer to the Rapid point of care test. We have included this study also in the reference section.

This manuscript is a resubmission of an earlier submission. The following is a list of the peer review reports and author responses from that submission.

Round 1

Reviewer 1 Report

Present study compares the efficacy of NADAL Covid-19 antigen test compared to RT-PCR test. Authors claimed that immunochromatographic SARS-CoV-2 antigen testing is effective over RT-PCR in Covid-19 diagnosis. Overall data and presentation were poor and not providing strong evidence as the authors claim. I don’t recommend for manuscript for acceptance. 1. No appropriate control was used for validation. 2. Cross reactivity and specificity of the antigen test kit with other SARS coronaviruses was not given. 3. Given data is not enough to validate their claim that NADAL Covid-19 antigen test has high sensitivity than RT-PCR test.

Reviewer 2 Report

The study describes the use of NADAL Covid-19 antigen test as a tool to mitigate the spread of the deadly virus in LTC. Overall, the study is good and in need of current pandemic. Some suggestion to authors;

  1. Please delete the repetitive line in Abstract.
  2. Introduction: Please provide a reference for the line" the use of antigen tests are critical to curbing the viral spread..."
  3. Please use standard nomenclature for real-time PCR (RT-PCR) throughout the article.
  4. Methodology: please edit 37.5OC (degree sign needed)
  5. Will it be possible to include your Area under curve graph and RT-PCR Ct value results with your result. If not then at least put them as supplement results.
  6. Please define the elderly age group in methodology part.
  7. The table 2 representation and text explanation are not clear. Will it be possible to break your table 2 in two different parts? While the text says 613 additional Ag tests were done from seven facilities and table says in six LTCs. Please include the % representation inside the table only with addition of a column. Please see and correct.
  8. Please reframe the sentence: Quite a few employees(n=55) and residents (n=53) were tested more than once in the observed period.
  9. While to make better sense, please explain in terms of what percentage of test were detected through Ag-test vs RT-PCR of the line “The ad-ministered antigen tests were positive for nine employees and 26 residents. PCR testing detected another seven infected employees but six residents proved to be false positive in the antigen test. Only one of the employees who were tested two times was tested positive on antigen tests and another one on an rt-PCR test. Six residents tested two times were positive on antigen tests and another eight tested positives on rt-PCR tests. None of the employees or residents who were tested more than two times were positive on the antigen test or on the rt-PCR test.”
  10.  “Mayor”: I think you meant major. Also, the line is not clear “all but two LTCs prevented mayor uncontrolled outbreaks of COVID-19 with antigen test.” Please check.
  11. Conclusion: Will it be possible to put the viral load of false negative cases.  

Reviewer 3 Report

This work developed a test strip for SARS-CoV-2 antigen testing. This topic is important but the manuscript was not written in a proper manner. It looks like a technical report other than a scitific paper.

Major

  1. Title: delete “to control out-breaks of SARS-CoV-2” to avoid the biased expression. “On 28 October 2020, EU Commission Communication stated that “robust test-ing strategies and sufficient testing capacities are essential aspects of preparedness and response to COVID-19” [1].” The above-mentioned expression makes sense, but the title is misleading.
  2. Abstract: It is biased to express“Rapid antigen testing prevented the uncontrolled spread of the SARS Cov2 in six out of eight LTCs”. Authors should keep in mind that test can never control the COVID-19 spread. However, test serves well for early detection. According to the WHO or CDC, their document clarified the term of COVID-19 prevent or control.
  3. “Sensi-tivity (86.96%, 95%CI: 66.41%-97.23%) and specificity (88.24%, 95%CI: 80.35%-93.77%) of the NADAL Covid-19 antigen test were very high.” The “very high” is not proper due to the sensitivity and specificity is good but not excellent. Authors should pay more attention to not affecting the actual evaluation of their results.
  4. The logic is not good enough. Authors should clarify the existing challenges in detecting COVID-19 and how they find a novel strategy to address the challengs. However, most sentences make no sense because they described massive information not related to the topic. Authors are suggested to focus on what they will do.
  5. Materials and Methods. The details of the experiment, for example, sample collection, sample preparation, result reading, should be listed in detail. They are laking the logic.
  6. It is suggested to discuss the scitific aspect of the experiments in detail and compare the current results with previous reports. For example, how to avoid false negative, which is not acceptale in practice.